# The RECQL helicase prevents replication fork collapse during replication stress

Bente Benedict[1], Marit AE van Bueren[1,*], Frank PA van Gemert[1,*], Cor Lieftink[2], Sergi Guerrero Llobet[3], Marcel ATM van Vugt[3], Roderick L Beijersbergen[2], Hein te Riele[1]

Most tumors lack the G1/S phase checkpoint and are insensitive to antigrowth signals. Loss of G1/S control can severely perturb DNA replication as revealed by slow replication fork progression and frequent replication fork stalling. Cancer cells may thus rely on specific pathways that mitigate the deleterious consequences of replication stress. To identify vulnerabilities of cells suffering from replication stress, we performed an shRNA-based genetic screen. We report that the RECQL helicase is specifically essential in replication stress conditions and protects stalled replication forks against MRE11-dependent double strand break (DSB) formation. In line with these findings, knockdown of RECQL in different cancer cells increased the level of DNA DSBs. Thus, RECQL plays a critical role in sustaining DNA synthesis under conditions of replication stress and as such may represent a target for cancer therapy.

## Introduction

DNA replication stress is increasingly being recognized as a hallmark of cancer. Uncontrolled proliferation driven by activation of oncogenes or loss of tumor suppressor genes often perturbs DNA replication and causes replication forks to decelerate, stall, or collapse (Zeman & Cimprich, 2014; Rickman & Smogorzewska, 2019). Replication stress may lead to the accumulation of single-strand DNA (ssDNA), DNA double-strand breaks (DSBs), and underreplicated DNA, which can activate the DNA damage response (DDR) to trigger cell cycle arrest or apoptosis. Previous studies have shown that tumors with high levels of replication stress were specifically sensitive to inhibition of certain DDR and cell cycle kinases, including WEE1, CHK1, ATM, and ATR (Weber & Ryan, 2015). Inhibition of these kinases aggravated replication stress and induced DNA damage that can eventually lead to apoptosis (Forment & O'Connor, 2018). Therefore, highly selective inhibitors are currently in clinical development (Mei et al, 2019). In addition, inhibition of the DDR may enhance the efficacy of traditional chemotherapeutic drugs that interfere with DNA replication by damaging the DNA or depleting the intracellular nucleotide pool (Lecona & Fernandez-Capetillo, 2018). These findings may point to a potential vulnerability of cancer cells: the dependence on mechanisms that sustain proliferation in the presence of replication stress. Identifying such mechanisms may yield novel therapeutic targets for cancer treatment (Zhang et al, 2016; Forment & O'Connor, 2018).

Perturbed DNA replication in cancer cells likely is a direct consequence of defective cell cycle control. In normal cells, cell cycle progression is regulated by checkpoints. The first checkpoint a newly formed cycling cell encounters is the G1/S checkpoint, which is activated by growth-restrictive conditions. The retinoblastoma (Rb) family members remain hypo-phosphorylated and stay bound to E2F transcription factor proteins. Therefore, E2F-dependent transcription of genes essential for S phase is inhibited and cells will arrest in the G1 phase (Burkhart & Sage, 2008; Bertoli et al, 2013). Cancer cells often lack the G1/S checkpoint and can, therefore, prematurely initiate DNA synthesis in growth-restricting conditions.

To mimic this situation and study the consequences of a defective G1/S checkpoint, we previously disrupted the three *Rb* genes in MEFs. Note that absence of all *Rb* genes is equivalent to overexpression of *cyclin D*, the second most frequently amplified locus in human cancers (Hosokawa & Arnold, 1998; Beroukhim et al, 2010). When cultured in the absence of mitogens, these *Rb*-deficient cells could still enter S phase and start synthesizing DNA; however, they suffered from severe replication stress that activated the DDR checkpoint and arrested cells in a G2-like state (van Harn et al, 2010; Benedict et al, 2018). Additional disruption of *Tp53* allowed cells to proliferate mitogen-independently, although DNA replication speed was still severely reduced. We therefore reasoned that proliferation of these cells relies on mechanisms that mitigate the deleterious consequences of replication stress.

In the present study, we sought to identify mechanisms that cancer cells need to deal with replication stress. We performed an

[1]Division of Tumor Biology and Immunology, The Netherlands Cancer Institute, Amsterdam, The Netherlands  [2]Division of Molecular Carcinogenesis, Robotics and Screening Center, The Netherlands Cancer Institute, Amsterdam, The Netherlands  [3]Department of Medical Oncology, Cancer Research Center Groningen, University Medical Center Groningen, Groningen, The Netherlands

Correspondence: h.t.riele@nki.nl
*Marit AE van Bueren and Frank PA van Gemert contributed equally to this work

shRNA-based screen interrogating the DDR subset of the genome to identify specific genes that are essential for mitogen-independent proliferation. We report that in addition to several DDR kinases, the helicase RECQL is essential to maintain proliferation in replication stress conditions. RECQL (also known as RECQL1 or RECQ1) belongs to the RECQ family of helicases, enzymes with 3′-5′ directed DNA unwinding capacity that help maintaining genomic integrity by processing aberrant DNA structures that arise during DNA replication (Bernstein et al, 2010; Croteau et al, 2014). In humans, RECQL is the most abundant member of the family. Previous studies have suggested that the RECQL helicase promotes the restart of replication forks that have reversed upon DNA topoisomerase I inhibition by using its ATPase and branch-migrating activities (Berti et al, 2013; Chappidi et al, 2020). However, whether RECQL is also important during other types of replication stress conditions is not clear yet.

Here, we demonstrate that RECQL activity protects cells experiencing replication stress against MRE11-dependent DNA DSB formation. Concordantly, inactivation of RECQL in human tumor cell lines of various origins induced DNA DSB formation. Therefore, RECQL represents a vulnerability of cells suffering from replication stress and could be a candidate for targeted anticancer therapy.

# Results

## Replication stress in mitogen-starved G1/S checkpoint-defective cells

We previously engineered MEFs lacking the G1/S checkpoint. These so-called TKO-Bcl2 MEFs, which are devoid of the retinoblastoma genes *Rb1*, *Rbl1* (*p107*), and *Rbl2* (*p130*) and overexpress the anti-apoptotic gene *Bcl2*, can start DNA replication in the absence of serum (Foijer et al, 2005; van Harn et al, 2010; Benedict et al, 2018). However, these cells experienced replication stress, revealed as slow replication fork progression, decreased levels of origin firing and high amounts of DNA DSBs, which activated the DDR and arrested cells in a G2-like state. Concomitant inactivation of *p53* restored the level of origin firing, reduced DNA breakage, and allowed proliferation in the absence of serum. Although these TKO-Bcl2-p53KO (TBP) MEFs were able to proliferate in the absence of growth factors, we previously showed that they still experienced replication problems as judged by low replication fork speed (Benedict et al, 2018).

To further validate the presence of replication stress, we studied the response of these cells to small-molecule inhibitors of the DDR. Inhibition of ATR, ATM, or DNA-PK or their downstream targets strongly suppressed proliferation of serum-starved TBP MEFs, whereas treatment did not or only marginally affect proliferation in the presence of mitogens (Fig S1A–F). In addition, treatment with the ATR inhibitor VE821 or with UCN01, an inhibitor of its downstream effector kinase CHK1, induced apoptosis specifically in replication stress conditions (Fig S1G and H). Tracking the cell cycle in individual cells using the FUCCI system revealed that mitogen-deprived TBP MEFs treated with the ATR or CHK1 inhibitors mostly died in the S or G2 phase (Fig S2A–E). Thus, TBP MEFs cultured in the absence of serum suffered from replication stress, evidenced by reduced replication fork speed and strong dependence on the ATR and CHK1 checkpoint kinases.

## An shRNA screen to identify genes essential in replication stress conditions

To identify mechanisms that are essential for mitogen-independent proliferation of TBP cells, we performed an shRNA-based drop-out screen. To this aim, we assembled a sub-library of the Sigma-Aldrich Mission TRC lentiviral shRNA library that is composed of shRNA vectors targeting 392 genes implicated in the DDR, with each gene targeted by five different shRNAs. TBP MEFs were infected with the library and cultured in the presence or absence of serum. ShRNAs that were depleted from the serum-deprived population likely target genes that are essential for proliferation in replication stress conditions (Figs 1A and S3A). After 10–12 cell divisions, we isolated genomic DNA, recovered shRNA inserts by PCR amplification and performed next generation sequencing to quantify the shRNA sequences. Individual replicates highly correlated and clustered together (Fig S3B and C). As we set out to identify genes essential in replication stress conditions but not in normal conditions, we only selected genes that were targeted by at least two shRNAs that each showed at least a twofold reduction upon culturing in the absence of serum compared with culturing in the presence of serum (Fig 1B). Several genes involved in DDR were found to be critical in mitogen-starved cells, such as *Clspn* and *Chek1*. Knockdown (KD) of *Chek1* in TBP cells in a separate experiment (Fig 1C and D) indeed inhibited growth in the absence of serum, whereas cells proliferated normally in the presence of serum (Figs 1E and S3D), which is consistent with the effects of pharmacological inhibition of CHK1 (Fig S1C).

## The RECQL helicase is essential for proliferation in replication stress conditions and prevents DNA damage

In addition, different shRNAs targeting the *Recql* gene significantly and reproducibly dropped out in serum-deprived conditions (Fig 1B). In a validation experiment, KD of *Recql* in TBP MEFs (TBP-RecqlKD MEFs, Fig 2A and B) did not affect proliferation in normal conditions (Fig 2C), whereas in the absence of mitogens, *Recql*KD inhibited proliferation of TBP MEFs and induced apoptosis (Fig 2D and E). Note that proliferation of the control TBP MEFs was slower in the absence of serum because of the presence of replication stress as previously described (Benedict et al, 2018). These results were validated in two additional TBP-RecqlKD lines generated with newly designed *Recql* KD vectors (Fig S4A–D). RECQL is a member of the family of RECQ ATP-dependent DNA-unwinding enzymes that play a central role in maintaining chromosome stability (Bernstein et al, 2010; Croteau et al, 2014). RECQL is the most abundantly expressed helicase of the RECQ helicase family, but its biological significance is not properly understood. RECQL is highly expressed in various cancer types and tumor cell lines (Mendoza-Maldonado et al, 2011; Sharma, 2014; Viziteu et al, 2017). Furthermore, RECQL has been suggested as target for anticancer drugs as its inhibition reduced proliferation of cancer cell lines (Sharma & Brosh, 2007).

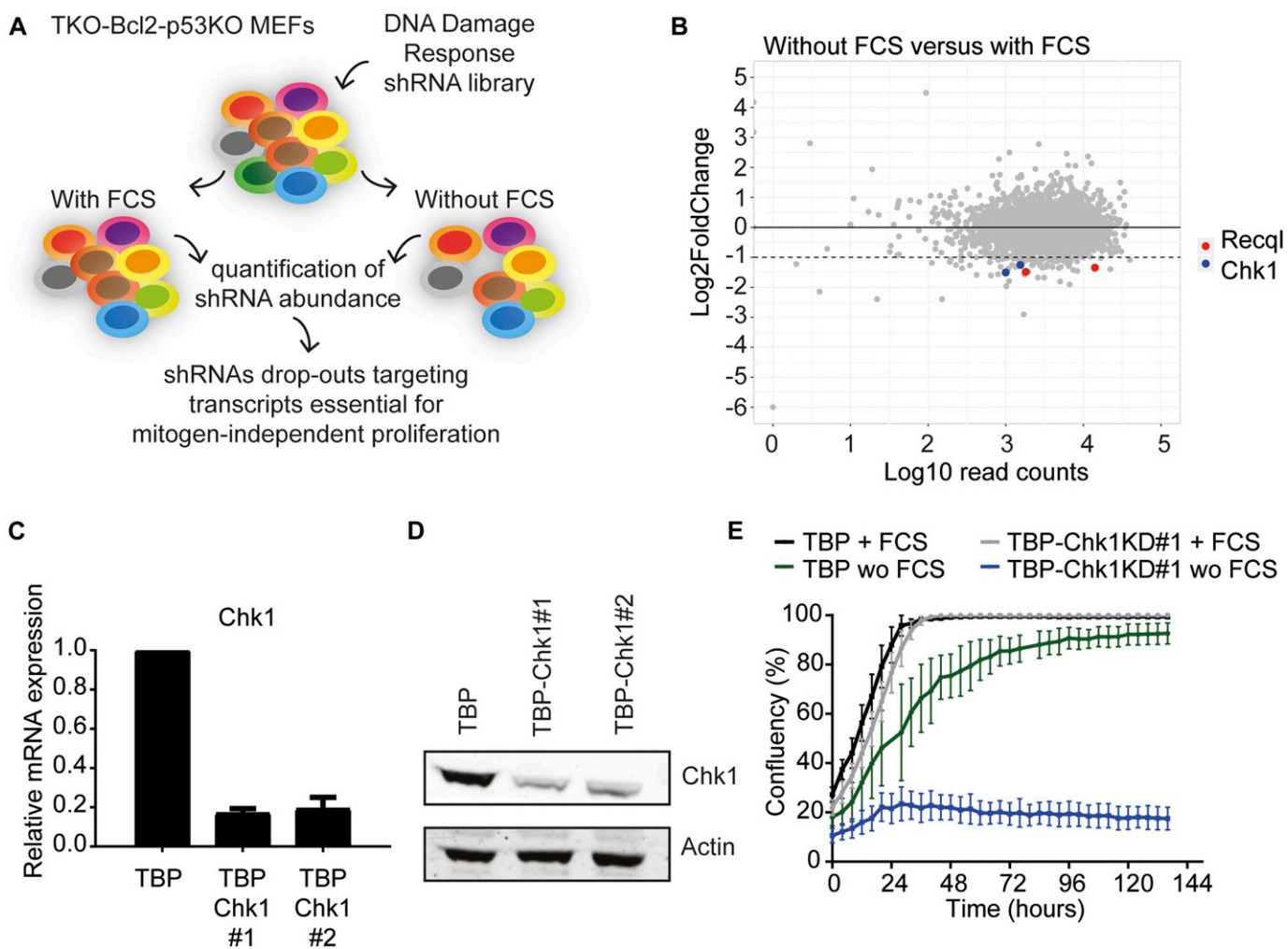

**Figure 1.   shRNA-based screen to identify genes essential for proliferation in replication stress conditions.**
**(A)** Schematic outline of the shRNA screen. TBP MEFs were infected with the lentiviral DNA damage response shRNA library and cultured in the absence or presence of FCS. ShRNA inserts were subsequently recovered by PCR and analyzed by next generation sequencing. ShRNAs that were depleted in the absence of serum likely target genes essential for proliferation in replication stress conditions. **(B)** The shRNA screen identified expression of CHK1 and RECQL essential for proliferation in replication stress conditions. The x-axis shows the average number of sequencing reads at the start point. The y-axis depicts the fold change in abundance of shRNAs in the cells cultured without FCS versus with FCS. **(C)** CHK1 expression levels in TBP, TBP-Chk1KD#1 and TBP-Chk1#2 MEFs measured by qRT-PCR. Error bars show SD of two independent experiments. **(D)** CHK1 protein levels in TBP, TBP-Chk1KD#1 and TBP-Chk1#2 MEFs. Anti-actin was used as loading control. **(E)** Growth curve of TBP MEFs with FCS (black) and without FCS (green), TBP-Chk1KD#1 MEFs with FCS (grey), and without FCS (blue) measured using the IncuCyte. Error bars show SD.

Using the FUCCI system, we observed no significant difference in duration of G1 and S/G2 phases between TBP-RecqlKD and TBP MEFs, either with or without FCS (Fig S4E). To determine if KD of *Recql* increased the amount of DNA DSBs, the neutral comet assay was used. In this assay, the tail moment is a measure for the amount of DNA DSBs. TBP-RecqlKD MEFs showed a clear increase in tail moment after 2, 3, 4 and 7 d of mitogen starvation (Figs 2F and S5A, B, and E). However, as also the fraction of apoptotic cells in serum-starved TBP-Recql MEFs was higher than in TBP MEFs (Fig 2E), could the comet assay have merely detected apoptotic cells? Several observations argue against this possibility. First, after 6 d of serum starvation, the fraction of apoptotic TBP and TBP-RecqlKD MEFs became similar (Fig 2E). However, at the 7 d time point, the comet assay did not detect DSBs in TBP MEFs, whereas it did in TBP-

RecqlKD MEFs (Fig 2F). Second, we cultured TBP-RecqlKD MEFs for 4 d without serum, but now in the presence of the pan-caspase inhibitor QvD, which effectively suppressed apoptosis (Fig S5C). In this experiment, the amount of DNA DSBs was again exclusively increased in TBP-RecqlKD MEFs (Fig S5C and D). Third, we reasoned that DNA breakage causes replicative arrest, and therefore, we labelled cells for 30 min with the thymidine analogue CldU and quantified γH2AX foci per nucleus in CldU-negative cells. Whereas TBP MEFs did not show an increase in γH2AX foci after 2 d of serum starvation, TBP-RecqlKD MEFs did show an increase in the amount of γH2AX foci (Fig 2G and H). Although not directly demonstrating DSBs, this experiment is indicative for activation of a DDR consistent with the increase in DNA DSBs as detected in the comet assay.

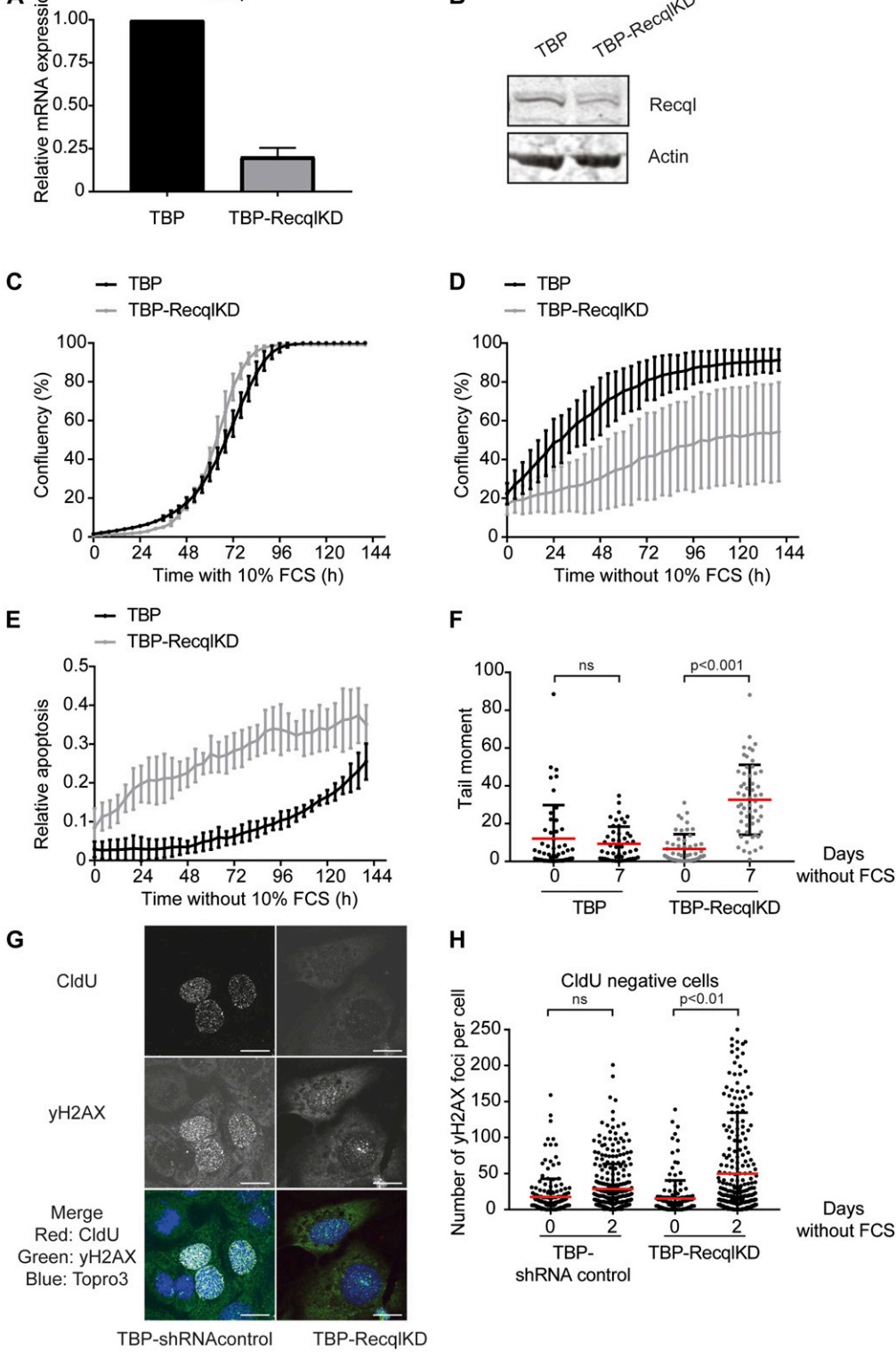

**Figure 2. RECQL is essential for proliferation in replication stress conditions.**
**(A)** RECQL expression levels in TBP and TBP-RecqlKD MEFs measured by qRT-PCR. Error bars show SD of two independent experiments. **(B)** RECQL protein levels in TBP and TBP-RecqlKD MEFs. Anti-actin was used as loading control. **(C, D)** Growth curves of TBP (black) and TBP-RecqlKD (grey) MEFs cultured in the presence (C) and absence of FCS (D) measured with the IncuCyte. **(E)** Relative apoptosis of TBP (black) and TBP-RecqlKD (grey) MEFs cultured without FCS. Apoptosis was measured by fluorescent signal upon caspase 3 cleavage and normalized to cell confluency. For (C, D, E), error bars show SD. **(F)** Tail moments of TBP (black) and TBP-RecqlKD (grey) MEFs cultured with FCS or for 7 d without FCS. For each condition, more than 50 cells were analyzed using CASP software. SDs are plotted in black and red bars denote the mean. Significance is indicated (one-way ANOVA nonparametric Kruskal–Wallis test). **(G)** Immunofluorescence images of γH2AX and CldU foci in TBP-shRNA control and TBP-RecqlKD MEFs after 2 d of serum starvation. DNA was labelled with Topro3. In the merged picture, DNA is blue, γH2AX is green and CldU is red. Colocalization of γH2AX and CldU is seen as yellow foci. Scale bar = 12 μm. **(H)** Quantification of the number of γH2AX foci per nucleus in CldU-negative TBP-shRNA control and TBP-RecqlKD MEFs in the presence and absence of serum. Mean is indicated in red. Results of three independent experiments are pooled and significance is indicated (*t* test).

We, therefore, conclude that expression of the helicase RECQL is essential to maintain proliferation in replication stress conditions and to prevent the accumulation of DNA DSBs.

## RECQL is not essential for the repair of broken replication forks

To study whether RECQL depletion causes increased DNA breakage or reduced DSB repair, we used hydroxyurea (HU) to induce replication stress. HU causes an unbalance in the nucleotide pool and thereby slows down DNA synthesis, causing replication fork stalling and collapse (Bianchi et al, 1986; Koç et al, 2004; Singh & Xu, 2016). Previously, we and others have observed that treatment with a low dose of HU (300 $\mu$M) causes replication fork stalling, but not replication fork collapse, that is, DNA DSBs (Benedict et al, 2018; Roy et al, 2018). In contrast, high levels of replication stress induced by treatment with a high dose of HU (2 mM) caused DNA DSB formation (Coyle & Strauss, 1970; Walker et al, 1977, Benedict et al, 2020, 2018). To determine whether RECQL impacts the formation of replication-associated DSBs, we cultured TBP and TBP-RecqlKD MEFs in the presence of FCS and treated the cells with 300 $\mu$M or 2 mM HU for 1 h and performed neutral comet assays (Fig 3A). Strikingly, whereas both cell lines showed an increase in DSBs after treatment with 2 mM HU, all three independently generated TBP-RecqlKD lines, but not *Recql* wild-type MEFs, showed an increase in DNA DSBs after the 1 h treatment with 300 $\mu$M HU levels (Figs 3B and S6A). Also treatment with 300 $\mu$M HU for 0.5, 2, 3, and 5 h increased the level of DNA DSBs in RecqlKD MEFs but not in control cells (Fig S6B and C).

To study whether RECQL affects DNA DSB repair, we performed neutral comet assays immediately after 1 h of 300 $\mu$M HU treatment or 30 min after wash away of the HU (Fig 3C). Treatment with 300 $\mu$M HU induced DNA DSB specifically in the TBP-RecqlKD MEFs, and these DSBs were repaired within 30 min after HU was washed away (Fig 3D). In line with 1 h 300 $\mu$M HU treatment, both cell lines repaired the DNA DSBs induced with 2 mM HU treatment within 30 min after wash away of the HU (Fig 3E) and this repair was dependent on RAD51 but not on DNA-PK (Fig S7A and B). Altogether, we suggest that RECQL prevents the formation but is not involved in the repair of HU-induced DNA DSBs that likely occur at the replication fork.

## RECQL protects stalled replication forks against MRE11-dependent DNA DSB formation

Replication fork breakage is a process that has been attributed to the activity of structure-specific nucleases, such as MRE11 (Pasero & Vindigni, 2017). To investigate whether the increase in DNA DSBs in the RECQL KD MEFs is effectuated by MRE11 nuclease activity, we used the MRE11 inhibitor Mirin (Dupré et al, 2008). MEFs were treated with 50 $\mu$M Mirin overnight and subsequently with 300 $\mu$M HU for 1 h. DNA breakage was assessed by the neutral comet assay. Whereas the amount of DSBs was increased in TBP-RecqlKD MEFs after 1 h of 300 $\mu$M HU treatment, the presence of Mirin significantly reduced DSB induction (Fig 3F). This suggests that the DSBs formed in TBP-RecqlKD MEFs could—at least in part—be attributed to MRE11 activity. Similarly, inhibition of MRE11 by Mirin treatment reduced DNA DSBs in TBP-RecqlKD MEFs cultured for 7 d without mitogens (Fig 3G). Moreover, Mirin treatment rescued the proliferation defect

of TBP-RecqlKD MEFs cultured in the absence of serum (Fig 3H). To verify that the observed effects of Mirin were due to inhibition of MRE11, we knocked down MRE11 using siRNAs in TBP and TBP-RecqlKD MEFs (Fig S8A and B). KD of MRE11 in TBP-RecqlKD MEFs reduced induction of DNA DSBs by a 1 h treatment with 300 $\mu$M HU (Fig S8C), suggesting that MRE11 is responsible for the DNA DSBs generated in the absence of RECQL.

Notably, inhibition of MRE11 by Mirin also suppressed DNA DSB formation in TBP MEFs treated with 2 mM of HU (Fig S8D). This indicates that MRE11 is involved in the formation of DNA breaks upon the presence of high levels of replication stress, and this MRE11 activity is apparently not directly inhibited by RECQL. Instead, these results suggest that RECQL reduces the vulnerability of stalled replication forks to MRE11-mediated DSB formation under low levels of replication stress.

MRE11 has previously been implicated in nascent strand degradation (Schlacher et al, 2011). A similar activity has been ascribed to another nuclease, DNA2, which is also suppressed by RECQL (Thangavel et al, 2015). When we reduced DNA2 expression in TBP-RecqlKD MEFs, DNA DSBs were still formed after 1 h of treatment with 300 $\mu$M HU (Fig S8E–G), arguing against a fork destabilizing activity that is shared by MRE11 and DNA2. Another possibility is that MRE11 is not directly causal to DNA DSB formation but precedes the activity of another fork nuclease such as the endonuclease MUS81. However, reduced expression of MUS81 did not rescue the DNA DSB formation in TBP-RecqlKD MEFs after treatment with 300 $\mu$M HU (Fig S8H–J).

## RECQL is involved in fork restart at broken, but not stalled replication forks

Previous studies suggested a role for RECQL in restarting regressed replication forks after topoisomerase I inhibition (Berti et al, 2013; Chappidi et al, 2020). To study if RECQL is also required for resuming DNA synthesis after 300 $\mu$M HU treatment, we quantified replication fork restart ability of TBP-RecqlKD MEFs using the DNA fiber assay. In a first experiment, cells were labelled for 15 min with the thymidine analogue CldU and then treated with 300 $\mu$M HU to induce replication fork stalling; after 1 h, HU was washed away and subsequently cells were labelled for 30 min with the thymidine analogue IdU. Using this protocol, stalled replication forks will be visible as red-only tracks, whereas restarted replication forks are visible as red–green or green–red–green tracks (Fig 4A). After HU washout, ~70% of the stalled replication forks were able to restart in TBP MEFs. However, upon KD of *Recql*, only 40% of the stalled replication forks had resumed DNA synthesis (Fig 4B). Possibly though, in the absence of RECQL, 300 $\mu$M HU induces DNA DSBs that first need to be resolved before DNA synthesis can resume, which may have delayed fork restart in the absence of RECQL. To test this possibility, we analyzed replication fork restart after treatment with 2 mM HU, which induced DSBs in both TBP and TBP-RecqlKD MEFs. We observed that in the absence of RECQL the percentage of restarted replication forks was still decreased twofold with respect to TBP MEFs (Fig 4C) similar to 300 $\mu$M HU treatment. This suggests that RECQL is involved in resuming DNA synthesis after a broken replication fork is repaired.

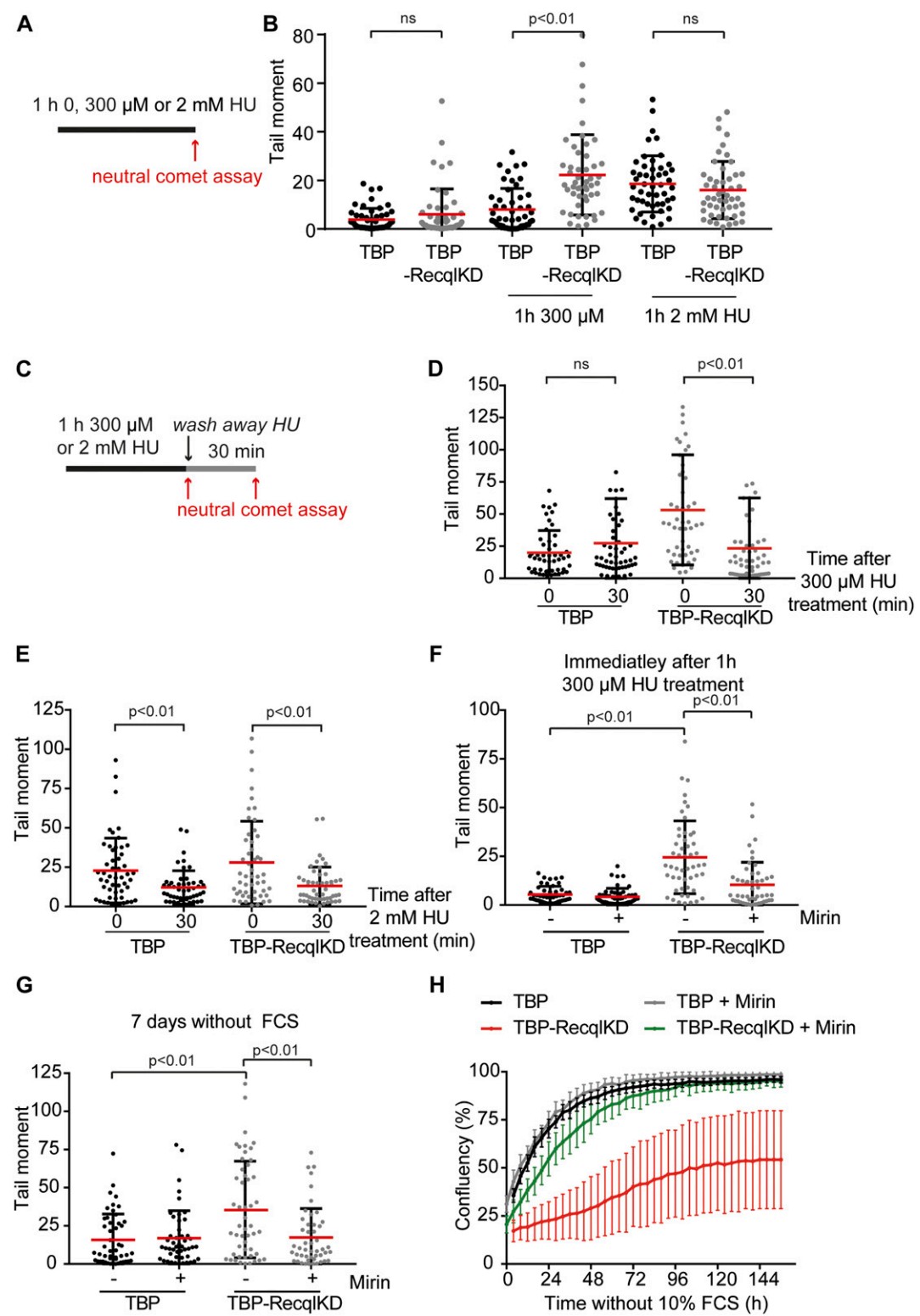

**Figure 3. RECQL protects stalled replication forks against MRE11-dependent DNA double-strand break formation.**
**(A)** Timing of the neutral comet assay procedure after 1 h with 0, 300 $\mu$M, and 2 mM HU. **(B)** Tail moments of TBP (black) and TBP-RecqlKD (grey) MEFs without HU treatment or immediately after 1 h 300 $\mu$M or 2 mM HU treatment. **(C)** Timing of the neutral comet assay procedure after 1 h with 300 $\mu$M or 2 mM HU or 30 min after washing away HU. **(D)** Tail moments of TBP (black) and TBP-RecqlKD (grey) MEFs immediately after 1 h 300 $\mu$M HU treatment or 30 min after washing away HU. **(E)** Tail moments of TBP (black) and TBP-RecqlKD (grey) MEFs immediately after 1 h 2 mM HU treatment or 30 min after washing away HU. **(F)** Tail moments of TBP (black) and TBP-RecqlKD (grey) MEFs after 1 h 300 $\mu$M HU treatment in the presence or absence of 50 $\mu$M Mirin. We previously showed that Mirin was active in MEFs and reduced repair of double strand

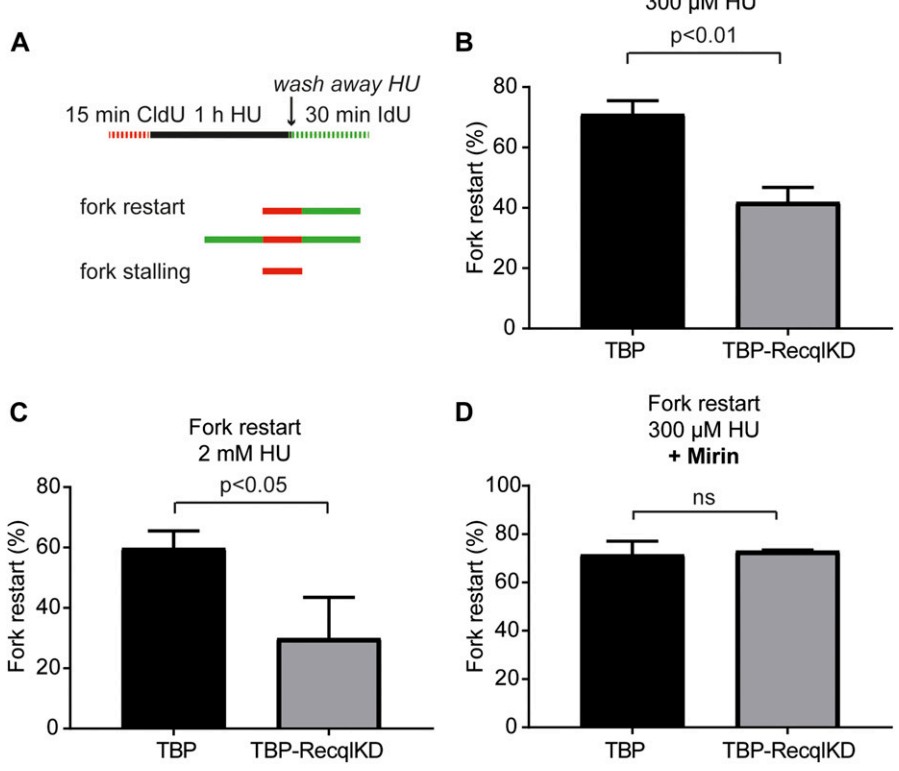

Figure 4. **RECQL is involved in fork restart at broken, but not stalled replication forks.**
**(A)** Schematic overview of the DNA fiber assay procedure to measure replication fork restart. Cells were labelled with CldU, treated with HU for 1 h, washed and labelled with IdU as indicated.
**(B)** Quantification of fork stalling in TBP (black) and TBP-RecqlKD MEFs (grey) after 300 $\mu$M HU treatment.
**(C)** Quantification of fork stalling in TBP (black) and TBP-RecqlKD MEFs (grey) after 2 mM HU treatment.
**(D)** Quantification of fork stalling in TBP (black) and TBP-RecqlKD MEFs (grey) after 300 $\mu$M HU treatment in the presence of 50 $\mu$M Mirin. For (B, C, D), at least 300 different fiber types were counted. Error bars show SDs. Significant differences between three independent experiments are indicated ($t$ test).

To address the question whether RECQL is also necessary for resuming DNA synthesis at a stalled replication fork, we treated cells again with 300 $\mu$M HU but now in the presence of Mirin to avoid DNA breaks in RECQL-depleted cells. In the presence of Mirin, TBP-RecqlKD MEFs were able to restart stalled replication forks as effective as TBP MEFs (Fig 4D). This indicates that when replication forks are only stalled, but not broken, RECQL is not essential for replication fork restart.

### RECQL prevents DSB formation in cells suffering from c-MYC–induced replication stress

In previous experiments, we studied replication stress induced by loss of the G1/S phase checkpoint or treatment with HU. Another cause of replication stress is the expression of oncogenes such as *c-Myc* (Gabay et al, 2014; Kuzyk & Mai, 2014). To study if RECQL is also involved in mitigating oncogene-induced replication stress, we used human RPE1 cells with a doxycycline-inducible *c-MYC* construct (Fig 5A). Upon overexpression of c-MYC, these RPE1 cells suffered from replication stress (Fig 5B–D). In contrast to loss of the G1/S phase checkpoint, where replication stress manifested as reduced fork speed accompanied by reduced origin firing (Benedict et al, 2018), untimely S phase initiation resulting from constitutive

oncogenic c-MYC signaling increased origin firing and decreased replication fork speed (Fig 5C and D). We created a KD of RECQL in the c-MYC–inducible RPE1 cells using an shRNA-targeting *Recql* (Fig 5E and F). Upon overexpression of c-MYC, KD of RECQL reduced proliferation earlier and induced slightly more cell death than wild-type cells (Fig 5G). Furthermore, KD of RECQL increased the level of DNA DSBs observed upon overexpression of c-MYC (Fig 5H), and formation of these DNA DSBs was dependent on the MRE11 nuclease (Fig 5I). This indicates that also upon induction of oncogene-induced replication stress, RECQL is important in preventing DNA DSB formation.

### RECQL prevents DSBs in human cancer cell lines

RECQL is highly expressed in various human cancer types and tumor cell lines (Mendoza-Maldonado et al, 2011; Sharma, 2014; Viziteu et al, 2017), and high expression levels are correlated with poor clinical prognosis (Li et al., 2006; Futami et al, 2010; Arai et al, 2011; Futami & Furuichi, 2014). Reduction of RECQL expression resulted in decreased proliferation of different cancer cell lines (Futami et al, 2010; Arai et al, 2011; Futami & Furuichi, 2014). Because our results suggested that RECQL protects against MRE11-dependent DNA DSB formation, we hypothesized that RECQL is essential in cancer cells

breaks induced by irradiation (Benedict et al, 2020). **(G)** Tail moments of TBP (black) and TBP-RecqlKD (grey) MEFs after 7 d without FCS in the presence or absence of 12.5 $\mu$M Mirin. **(H)** Growth curve of TBP MEFs (black), TBP MEFs with 12.5 $\mu$M Mirin (grey), TBP-RecqlKD MEFs (red), and TBP-RecqlKD MEFs with 12.5 $\mu$M Mirin (green). Error bars show SDs. For (B, D, E, F, G), more than 50 cells for each condition were analyzed using CASP software. SDs are plotted in black and red bars denote the mean. Significance is indicated (one-way ANOVA nonparametric Kruskal–Wallis test).

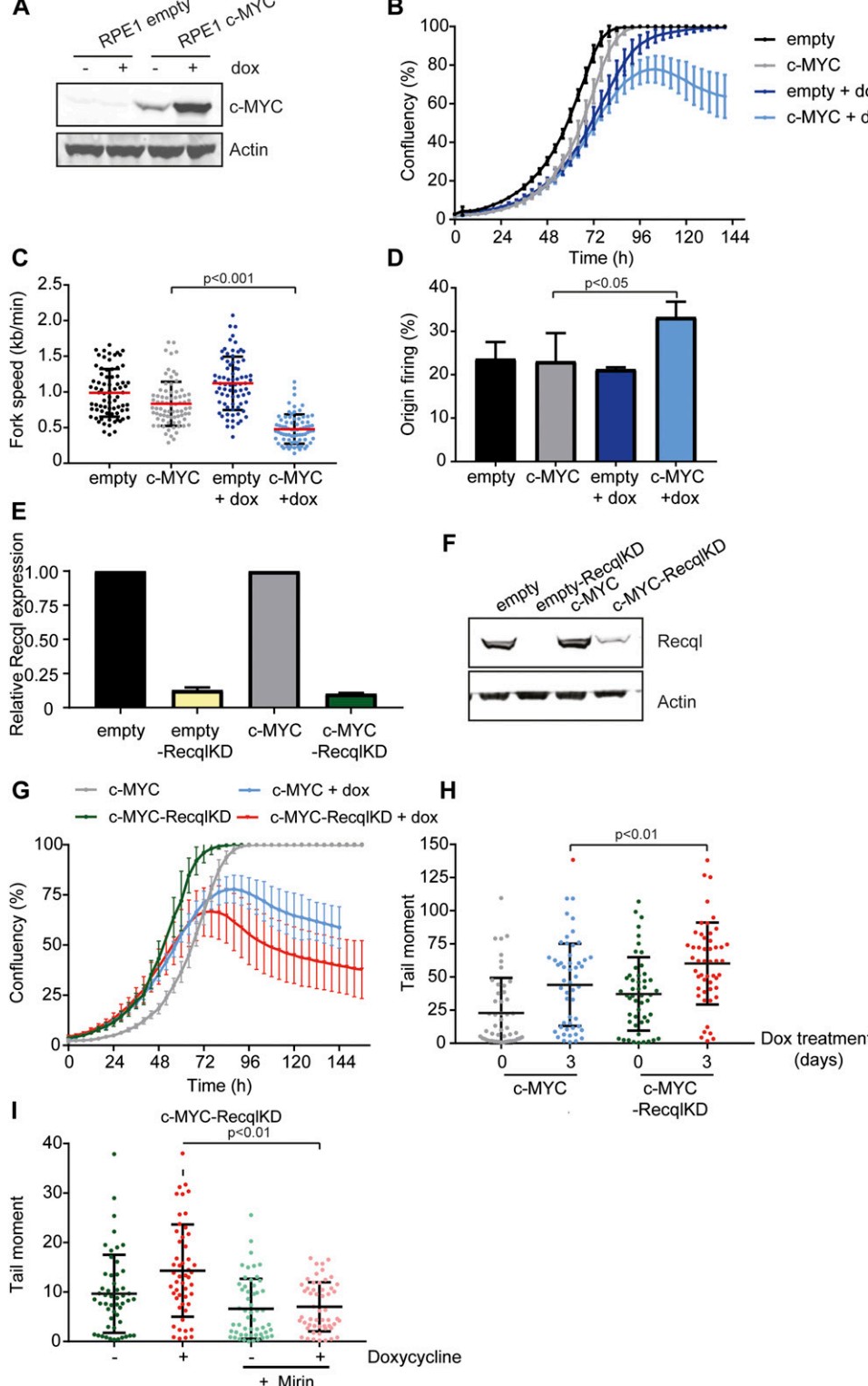

**Figure 5. RECQL prevents MRE11-dependent double strand break formation upon c-MYC–induced replication stress.**

**(A)** C-MYC protein levels in RPE1 cells with either an empty or *c-MYC*-inducible construct without doxycycline treatment or with 3 d of doxycycline treatment. Anti-actin was used as loading control. **(B)** Growth curves of RPE1 cells with empty and *c-MYC*-inducible construct cultured in the presence of doxycycline (black and grey, respectively) and absence of doxycycline (dark blue and light blue, respectively) measured with the IncuCyte. Error bars show SDs. **(C)** Replication fork speed of RPE1 cells with empty or c-MYC–inducible construct cultured in the presence or absence of doxycycline. Track lengths of at least 75 ongoing forks were measured with ImageJ. SD is plotted and red bars denote the mean. Significance is indicated (nonparametric Kruskal–Wallis test). **(D)** Quantification of origin firing in RPE1s with empty or *c-MYC*–inducible construct cultured in the presence or absence of doxycycline. First label and second label origins are shown as percentage of all labelled tracks. Error bars show SDs. Significant differences between three independent experiments are indicated (*t* test). **(E)** RECQL expression levels in empty, c-MYC, empty-RecqlKD, and c-Myc–RecqlKD RPE1s measured by qRT-PCR. Error bars show SDs of two independent experiments. **(F)** RECQL protein levels in empty construct and *c-MYC* doxycycline inducible RPE1s. Anti-actin was used as loading control. **(G)** Growth curves of c-MYC and c-MYC-RecqlKD RPE1s cultured in the absence (grey and green, respectively) and presence of doxycycline (blue and red, respectively) measured with the IncuCyte. Error bars show SDs. **(H)** Tail moments of c-MYC and c-MYC-RecqlKD RPE1s without doxycycline or with 3 d doxycycline treatment. **(I)** Tail moments of c-MYC-RecqlKD RPE1s in the presence (3 d) or absence of doxycycline in combination with or without 12.5 μM Mirin. For (H, I), more than 50 cells for each condition were analyzed using CASP software. SD and means are indicated in black. Significance is indicated (one-way ANOVA nonparametric Kruskal-Wallis test).

suffering from replication stress to prevent replication fork collapse. To test this hypothesis, we knocked down *RECQL* using siRNA in different human cancer cell lines and assessed the effect on DNA DSB formation. KD of *RECQL* (Fig 6A) increased the amount of DNA DSBs significantly in three of the six lines, whereas a slight increase was measured for the other three cancer cell lines (Fig 6B). To test if this effect was dependent on MRE11, we determined the effect of KD of RECQL on DNA DSB formation in the presence of Mirin for two cancer

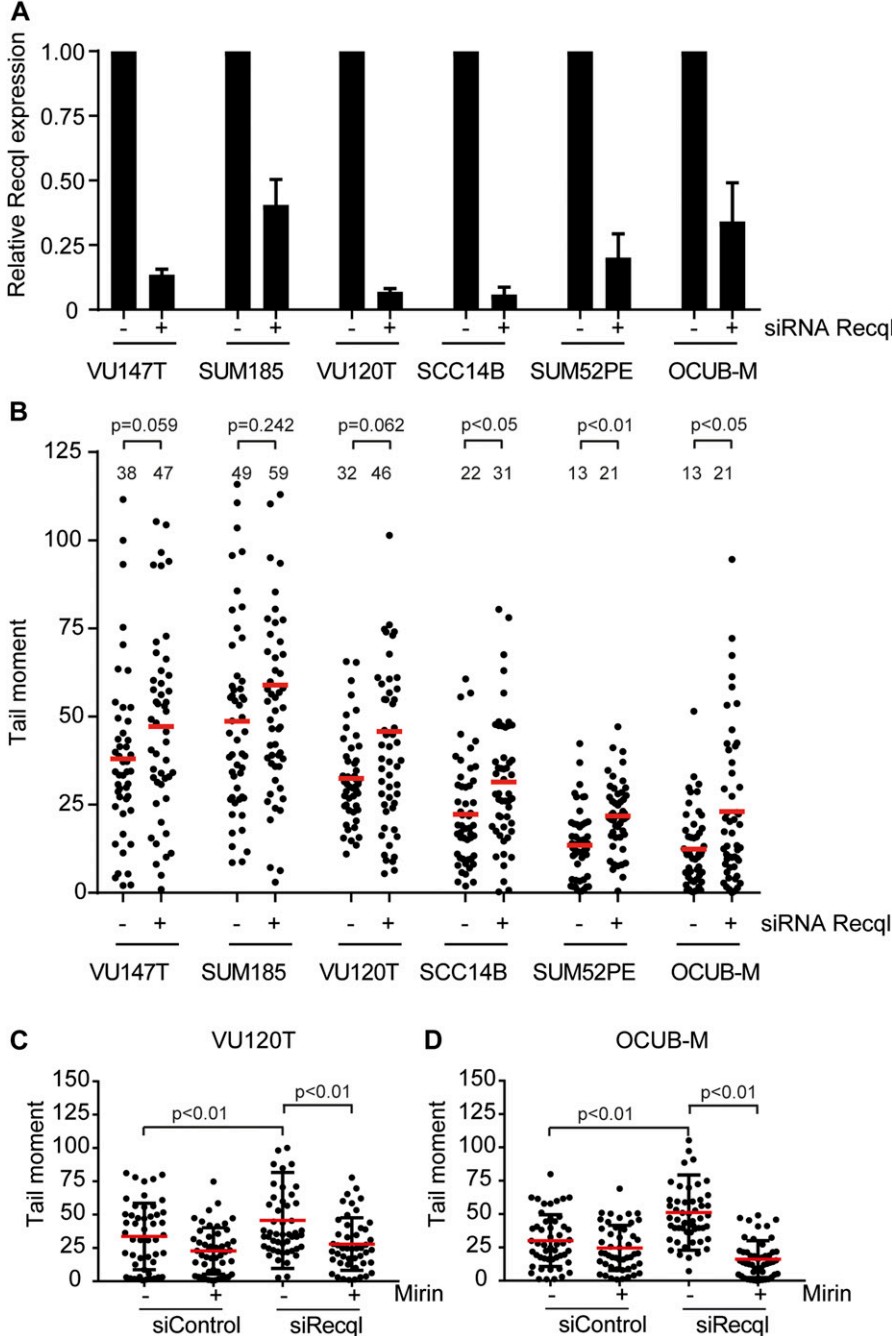

**Figure 6. RECQL prevents DNA double-strand break formation in human cancer cells.**
**(A)** RECQL expression levels of different cancer cell lines transfected with control and RECQL siRNAs measured by qRT-PCR. Error bars denote the SD of two independent experiments. **(B)** Tail moments of different cancer cell lines transfected with control and RECQL siRNAs. Mean of each condition is indicated by the red line and number. *P*-values are indicated above each cell line. **(C, D)** Tail moments of VU120T (C) and OCUB-M (D) cancer cells transfected with control and RECQL siRNAs and treated with and without 12.5 $\mu$M (C) and 6.25 $\mu$M (D) Mirin, respectively. SDs are plotted in black and red bars denote the mean. For (B, C, D), more than 50 cells for each condition were analyzed using CASP software. SDs are plotted in black and red bars denote the mean. Significance is indicated (one-way ANOVA nonparametric Kruskal–Wallis test).

cell lines. Similar to the results in the MEFs and RPE1 cells, inhibition of MRE11 using Mirin reduced the level of DNA DSBs (Fig 6C and D). This suggests that also in human cancer cells, RECQL prevents MRE11-dependent DNA DSB formation.

## Discussion

In most human tumors, the G1/S checkpoint is lost, for example, by loss of pRB or the CDK inhibitor p16[INK4A], or by overexpression of cyclin D1 (Ho & Dowdy, 2002) and insensitivity to antigrowth signals is a hallmark of tumor cells (Hanahan & Weinberg, 2000). Cells lacking the G1/S phase checkpoint can start synthesizing DNA under nonpermissive conditions, but this leads to DNA replication problems and DNA breakage, a state called "replication stress." Since untransformed cells have a functional G1/S phase checkpoint that precludes DNA replication in growth-inhibitory conditions, replication stress is unique for cancer cells, which may create an opportunity for therapeutic intervention (Zhang et al, 2016; Forment & O'Connor, 2018). To identify genes essential in replication stress

conditions, we used MEFs that lack the G1/S phase checkpoint because of ablation of the Rb-proteins, overexpress the anti-apoptotic gene *Bcl2*, and had lost p53 expression by gene disruption. In the absence of mitogenic signaling, these cells were able to proliferate, although they still suffered from replication stress as evidenced by reduced fork speed (Benedict et al, 2018) and dependence on the intra–S phase checkpoint, as shown here. Using an shRNA-based screen targeting DDR genes, we identified the RECQL helicase to be essential during replication stress induced by premature S phase entry. We show that RECQL prevents MRE11-dependent DNA breakage under conditions of perturbed replication. Although RECQL is not involved in DSB repair, our data suggest a role for RECQL in subsequent replication fork restart.

The biological function of RECQL is not yet properly understood, but previous biochemical studies have shown that RECQL preferably unwinds homologous recombination intermediates such as D-loops and Holliday junctions (LeRoy et al, 2005; Sharma et al, 2005; Popuri et al, 2008) and promotes branch migration in the 3′-5′ direction (Bugreev et al, 2008; Mazina et al, 2012). In contrast to its DNA unwinding function, RECQL can also catalyze the opposite reaction, the annealing of partially unwound DNA strands. It has been described that RECQL's assembly state determines the function; RECQL monomers or tight-binding dimers are associated with DNA unwinding, whereas oligomers consisting of five or six monomers are required for strand annealing (Muzzolini et al, 2007; Pike et al, 2015). Which activity is essential for the protection of stalled replication forks or the restart of broken forks is not clear.

When DNA synthesis stalls, the MCM helicase may still continue unwinding the parental DNA strands, creating a gap of ssDNA. This ssDNA is vulnerable to nucleolytic degradation. Previously, RECQL has been proposed to limit the activity of the DNA2 nuclease by preventing extensive nascent strand degradation (Thangavel et al, 2015). Here, we found an additional function of RECQL: suppressing MRE11-dependent DNA DSB formation in cells suffering from low levels of replication stress. We showed that high levels of replication stress generate forks that are vulnerable to MRE11 despite the presence of RECQL. This indicates that RECQL does not directly inhibit MRE11 activity. Instead, we speculate that under low levels of replication stress RECQL protects stalled forks from becoming sensitive to MRE11. Different mechanisms may underlie this activity. First, the helicase activity of RECQL could facilitate the resolution of secondary structures formed in the ssDNA at a stalled replication fork; the persistence of such structures may aggravate fork stalling and promote subsequent breakage by MRE11. Two other members of the RECQ helicase family, WRN and BLM, are also involved in the dissolution of ssDNA structures such as G-quadruplexes. It would be interesting to study whether RECQL binds specifically to DNA sequences that can form secondary structures. Second, RECQL may promote regression of replication forks through its annealing activity (Neelsen & Lopes, 2015), thereby reducing break-sensitive ssDNA stretches. Third, it has been speculated that RECQL is involved in the resolution of regressed replication forks (Berti et al, 2013; Chappidi et al, 2020). Possibly, RECQL could be involved in stabilizing and protecting the nascent DNA, similar to the function of BRCA2 and RAD51 at regressed replication forks (Schlacher et al, 2011, 2012; Zellweger et al, 2015; Rickman & Smogorzewska, 2019).

Rather than the protruded arm of a reversed replication fork, RECQL may protect the four-way junction point from nucleolytic cleavage, a function that has recently been ascribed to WRNIP1 (Porebski et al, 2019). Which of these mechanisms would create structures vulnerable to MRE11 activity? MRE11 has both endo- and exonuclease activity (Williams et al, 2008; Garcia et al, 2011). Because Mirin has been suggested to specifically inhibit the exonuclease activity of MRE11 (Dupré et al, 2008), we envision that in our cells, DNA breaks rely on the exonuclease activity of MRE11. It is unclear though whether such activity of MRE11 could promote breakage of long stretches of ssDNA. It is also possible that the activity of MRE11 is not directly causal to DNA DSB formation but precedes the activity of other fork processing nucleases. However, preliminary results suggest that the MUS81 and DNA2 nucleases are not involved. It would be interesting to study whether other nucleases are involved in MRE11-dependent DNA breakage.

Our data suggest that RECQL is involved in the restart of broken, but not stalled replication forks. A possibility is that the function of RECQL in processing Holliday junctions formed during repair of a broken replication fork is essential to resume DNA synthesis, whereas fork restart at a stalled replication fork does not involve the resolution of a Holliday junction. The latter may contrast to previous studies implicating RECQL in promoting restart of regressed replication forks induced by an inhibitor of topoisomerase I and a G4-DNA binding ligand (Berti et al, 2013; Chappidi et al, 2020). Probably, distinct ways of replication fork stalling induce different DNA structures that are dependent on different DNA remodeling enzymes.

The RECQL helicase family consists of five homologs: RECQL, WRN, BLM, RECQL4, and RECQL5. Mutations in WRN, BLM, or RECQL4 cause different genetic disorders, all associated with accelerated aging, genomic instability and tumorigenesis (Ellis et al, 1995; Yu et al, 1996; Kitao et al, 1999; Siitonen et al, 2003; Van Maldergem et al, 2006). The distinct clinical features and cellular phenotypes of these genetic disorders suggest that the different RECQ-helicases have non-overlapping functions. We hypothesize that each helicase can specifically solve an aberrant DNA structure and thereby promote proper DNA replication and chromosome segregation. Mutations in RECQL have not been linked to a syndrome, but loss of RECQL has been associated with breast cancer predisposition (Banerjee & Brosh, 2015; Cybulski et al, 2015; Sun et al, 2015), perhaps by inducing replication-associated DNA DSBs. On the contrary, it has also been shown that RECQL is highly expressed in various tumors, such as multiple myeloma, glioblastoma, ovarian cancer, and breast cancers (Mendoza-Maldonado et al, 2011; Sharma, 2014; Viziteu et al, 2017). Overexpression of RECQL may provide survival advantage to cancer cells by protecting stalled replication forks against breakage. Therefore, RECQL has been suggested as target for anticancer therapy as inhibition of RECQL reduced proliferation of cancer cell lines (Futami et al, 2010; Arai et al, 2011; Futami & Furuichi, 2014). Consistently, we showed that inhibition of RECQL induced DNA DSBs in cancer cells of different origins. All together, we showed that the RECQL helicase is an essential player in the protection of stalled replication forks and possibly the restart of broken replication forks, thereby representing a target for cancer therapy.

# Materials and Methods

## Cell culture

TKO-Bcl2-p53KO MEFs (TBP MEFs) were generated as previously described (Benedict et al, 2018). MEFs were cultured in GMEM (Invitrogen) + 10% FCS, 0.1 mM nonessential amino acids (Invitrogen), 1 mM sodium pyruvate (Invitrogen), 0.1 mM $\beta$-mercaptoethanol (Merck), and 100 $\mu$g/ml penicillin and streptomycin (Invitrogen). Breast cancer and HNSCC cells were cultured in DMEM (Gibco) + 10% FCS, 1 mM sodium pyruvate, and antibiotics. For serum starvation experiments, cells were seeded in the presence of serum and allowed to attach for 4 h. Subsequently, cells were washed with PBS and incubated in serum-free medium for the duration of the experiment. When indicated, cells were treated with hydroxyurea (Sigma-Aldrich), ATR inhibitor VE821 (Sigma-Aldrich), ATM inhibitor KU55933 (Sigma-Aldrich), Chk1 and Chk2 inhibitor AZD7762 (Abcam), Chk1 inhibitor UCN01, ATM and ATR inhibitor Caffeine (Sigma-Aldrich), Rad51 inhibitor B02 (Sigma-Aldrich), DNA-PK inhibitor NU7441 (Sigma-Aldrich), Mirin (Sigma-Aldrich), or the pan-caspase inhibitor QvD (Abcam).

hTERT-immortalized retinal pigment epithelium cells (RPE1s) were purchased from the American Type Culture Collection (CRL-4000), and cultured in DMEM F12 (Invitrogen) + 10% FCS and supplemented with 1 $\mu$g/ml doxycycline (Sigma-Aldrich) when indicated. To generate hTERT-RPE1-Tet-On cells, RPE1 cells were first transduced with pRetroX-Tet-On Advanced and selected with 800 $\mu$g/ml geneticin (G418 Sulfate; Thermo Fisher Scientific) for 7 d. Subsequently, hTERT-RPE1-Tet-On cells were transduced with pRetroX-Tight-Pur-c-Myc. To generate this construct, c-Myc was PCR amplified from MSCV-Myc-T58A-puro (plasmid #18773; Addgene) using the following primers: c-Myc forward: 5′-CGCGGCCGCCATGCCCCTCA ACGTTAGCTTC-3′ and c-Myc reverse: 5′-GATGAATTCTTACGCACAA-GAGTTCCG-3′. MSCV-Myc-T58A-puro was a kind gift from Dr. Scott Lowe (Hemann et al, 2005). The c-Myc PCR product was digested with NotI and EcoRI and ligated into the corresponding cloning sites of pRetroX-Tight-Pur. Cells transduced with pRetroX-Tight-Pur-c-Myc were selected for 2 d with 5 $\mu$g/ml puromycin dihydrochloride (Sigma-Aldrich).

## Constructs and lentiviral transfections

The pLKO.1 lentiviral vectors containing an shRNA against mouse RECQL are from the Mission mouse TRC v1.0 collection (Sigma-Aldrich): TBP-RecqlKD MEFs were made with vector TRCN0000115248, TBP-RecqlKD#2 with vector TRCN0000115249 and TBP-RecqlKD#3 MEFS with vector TRCN0000115250 (Table S1). The pLKO.1 lentiviral vector containing an shRNA (TRCN0000289591) against human RECQL is from the Mission human TRC v1.0 collection (Sigma-Aldrich; Table S1). The two constructs CSII-EF-MCS-mKO-hCdt1 (30/120) and CSII-EF-MCS-mAG-hGem (1/110) were a kind gift of A Miyawaki (Sakaue-Sawano et al, 2008). Lentivirus was generated by co-transfecting HEK293T cells with the vector of interest and the helper plasmids pMDLgpRRE, VSV-G, and pRSV-Rec with PEI. Retroviral vectors were co-transfected with the helper plasmids puMCV-Gag pol MMLV and pCMV VSVG. For both lenti- and retroviral transductions, 48 and 62 h posttransfection

viral supernatant was harvested, filtered, and MEFs were infected three times for 8–12 h in the presence of 8 $\mu$g/$\mu$l polybrene. MEFs were selected with 8 $\mu$g/ml puromycin and 10 $\mu$g/ml blasticidin.

## siRNA experiments

For KD experiments in the different cancer cell lines, synthetic siRNAs (Dharmacon) were transfected using RNAimax (Invitrogen). Non-targeting siRNA UAAGGCUAUGAAGAGAUAC; human siRECQL GCAAGGAGAUUUACUCGAA, GGCCAAAUCUAUAUUAUGA, GAAGAUUAUUGCA-CACUUU, AACAAGAGCUUAUUCAGAA; mouse siMRE11 GAGUAGAA-GACCUCGUAAA, GGUCCGACGUUUCCGAGAA, GAGAAAUACCAACGAAGAA, GAUCAAAGGUGGUCGGGCA; mouse siDNA2 GCGGAUCAUCAGCGACUUA, CGCCAGAUGCUGAUCGGUA, CGGAGGACUUCAUGCGUAA, GGAA-GAAGGCGGACGCUUU, and mouse MUS81 CAGCCGUGGUGGAUC-GAUA, CCAGAAAUGCUCCGAGAGU, CCUCAUCCUUGGAACGCAU, and GUGUGUGGACAUUGGCGAA.

## shRNA drop-out screen

The DDR subset of the Sigma-Aldrich Mission TRC v1.0 collection shRNAs in lentiviral vectors targeting 392 genes (Table S2), with five individual shRNAs for each gene, was used to infect TBP MEFs with a 1,000-fold coverage. MEFs were harvested at T = 0 or cultured in the presence or absence of 10% FCS for at least 12 cell divisions. gDNA was isolated using the DNAeasy blood & tissue kit (QIAGEN) following the manufacturer's protocol. ShRNAs were retrieved using Phusion High-Fidelity DNA Polymerase (Thermo Fisher Scientific) by a 2-step PCR protocol. First, 25 $\mu$g gDNA was used in the first round of PCR amplification with 10 $\mu$l 5× GC buffer, 1 $\mu$l 10 mM dNTP, 2.5 $\mu$l 10 $\mu$M PCR 1 indexed forward primer (Table S2, primers PCR1F1-9), 2.5 $\mu$l 10 $\mu$M PCR 1 indexed reverse primer (Table S2, primer PCR1R), 1.5 $\mu$l DMSO, 1 U Phusion Polymerase, and H$_2$O added up to 50 $\mu$l and run at (1) 98°C, 30 s; (2) 98°C, 30 s; (3) 60°C, 30 s; (4) 72°C, 1 min (steps 2, 3, and 4 for 20 times) and 72°C, 5 min. The product of PCR1 was used to setup the second PCR reaction similar to PCR1. In PCR2, primers containing P5 and P7 sequences (Table S2, primers PCR2F and PCR2R) were used. The abundance of the guideRNAs in the T = 0 and the serum-cultured and non–serum-cultured cells was determined by Illumina Next Generation Sequencing. For the analysis of the screen, a differential analysis on the shRNA level with DESeq2 was performed (Love et al, 2014). As hits, we selected genes for which at least two shRNAs had a log$_2$ fold change smaller than −1 and a FDR ≤ 0.1.

## Neutral comet assay

To asses DNA DSBs, neutral comet assays were performed as previously described (Olive & Banáth, 2006; Benedict et al, 2018).

## DNA fiber assay

DNA fiber assays were performed as previously described (Benedict et al, 2018). Briefly, cells were labelled with subsequently CldU and IdU as indicated per experiment. Cells were harvested and lysed using spreading buffer (200 mM Tris–Hcl, pH 7.4, 50 mM EDTA, and 0.5% SDS). Lysed cells were spread onto Superfrost Microscope slides (Menzel-Gläser) and fixed using 3:1 methanol: acidic acid for

10 min. To allow for immunodetection of DNA fibers, slides were incubated in 2.5 M HCl for 1 h and 15 min. Subsequently, DNA fibers were blocked using Blocking solution (PBS + 1% BSA + 0.1% Tween 20) for 1 h. To detect CldU and IdU, primary antibodies were rat-anti-BrdU (Clone BU1/75, 1:500; Abcam) and mouse anti-BrdU (clone B44, 1:750; Becton Dickinson Bioscience), incubated for 1 h. Subsequently, antibodies were fixed using 4% paraformaldehyde incubation for 10 min, and slides were incubated with the secondary antibodies goat-antimouse Alexa 488 and goat-antirat Alexa 555 (1:500; Invitrogen) for 1.5 h. To seal slides, slides were mounted with Vectashield. Images were taken with a Zeiss AxioObserver Z1 inverted microscope using a 63× objective equipped with a Hamamatsu ORCA AG Black and White CCD camera.

### Western blot

Cells were harvested and lysed in ELB buffer (150 mM NaCl; 50 mM Hepes, pH 7.5; 5 mM EDTA; 0.1% NP-40) containing protease inhibitors (Complete; Roche) for 30 min. Protein concentrations were determined using the BCA protein assay kit (Pierce). Primary antibody used to detect mouse RECQL was rabbit polyclonal anti-Recql (A300-450A; 1/1,000; Bethyl Laboratories). For detection of human RECQL, mouse monoclonal anti-Recql (A-9, Sc166388; 1/250; Santa Cruz) was used. For detection of actin, polyclonal goat anti-actin (I-19; 1/1,000; Santa Cruz) was used. Secondary antibodies used were IR Dye 800CW donkey antigoat IgG (LI-COR) and IR Dye 800CW goat-antimouse IgG (LI-COR).

### mRNA isolation, cDNA synthesis and qRT-PCR

Total RNA was isolated using the high pure RNA Isolation Kit (Roche). 1 $\mu$g of RNA was used to prepare cDNA by reverse transcriptase (Superscript II; Invitrogen) using random hexamer primers. cDNA was used as template for qRT-PCR in the presence of SYBR Green (Applied Bioscience). Relative amounts of cDNA were compared with actin as reference for total cDNA. For the detection of mouse RECQL, we used the 5′-CGGCTGACAGAAGGACATTT-3′ as forward primer and the 5′-CCGCCTCTCTGTGAGTTCCT-3′ as reverse primer. For the detection of human RECQL, we used 5′-GCGTCCGTTTCAGCTCTAACT-3′ and 5′-CGGCAGTTCCCTAACGCAT-3′ as forward primers and 5′-TTGCCCCGGCATCAGAATC-3′ and 5′-TCTTCAGTGTTTGAGGGCTTCT-3′ as reverse primers. For the detection of mouse ACTIN, we used 5′-TCCACCCGCGAGCACAGCTTCTTTG-3′ as forward primer and 5′-ACATGCCGGAGCCGTTGTCGACG-3′ as reverse primer. For the detection of human ACTIN, we used 5′-CATG-TACGTTGCTATCCAGGC-3′ as forward primer and 5′-CTCCTTAATGT-CACGCACGAT-3′ as reverse primer. For the detection of mouse MRE11 we used 5′-AAACTCGCTCTGTACGGCTTA-3′ and 5′-TCTGGCTAACCACC-CAAACC-3′ as forward primers and 5′-CAGATAACGAGGTCGATGAAGTC and 5′-GAGGATTCCCCAGTCGTTCC-3′ as reverse primers. For the detection of mouse DNA2 we used 5′-GATGCTGATCGGTACAATTCTCC-3′ and 5′-CTTGCCACAGATAATCGAGGAAG-3′ as forward primers and 5′-GGCTCAGATTCAAGCGATACAT-3′ and 5′-CTGGGATGGATGCGTCATCTC-3′ as reverse primers. For the detection of mouse, MUS81 we used 5′-TCGTGTTTCAAAAGGCATTGC-3′ and 5′-GTGGACATTGGCGAAACCAGA-3′ as forward primers and 5′-TCACCGCCTGATGCTAGGT-3′ and 5′-CTCCAACGTGTAGCTTGCGT-3′ as reverse primers.

### Growth curves, apoptosis and crystal violet assays

To monitor cell growth, the live cell imaging instrument IncuCyte ZOOM (Essen Bioscience) was used. Cells were plated in a 96-well Micro Greiner clear plate (Sigma-Aldrich) and imaged every 4 h with default software settings and a 10× objective. The IncuCyte software was used to quantify confluence from two non-overlapping brightfield images. To identify apoptotic cells, the IncuCyte Zoom instrument was used in combination with the Cell Player 96-well kinetic caspase-3/7 reagent (Essen Bioscience). The software was used to calculate mean green fluorescence for two non-overlapping images of each well. Green fluorescence was normalized to phase-contrast confluence to determine apoptosis. For crystal violet assays, the cells were cultured in six-well dishes and fixed using 4% formaldehyde for 10 min. Plates were washed using running tap water and stained using 1 ml 0.1% crystal violet for 30 min exactly.

### Time lapse microscopy

For FUCCI live cell imaging, MEFs were imaged using a Zeiss Axiovert 200M inverted microscope with a 37°C heated stage and images were captured with a 0.25 Ph1 Achroplan objective and Hamamatsu ORCA R2 Black and White CCD-camera. Images were taken every 30 min with a 20× objective and 1.6 optovar. Movies were analyzed using ImageJ.

### Immunofluorescence

CldU and yH2AX stainings were performed as previously described (Benedict et al, 2018). Shortly, cells were cultured on cover slides, labelled with 100 $\mu$M CldU for 30 min, washed with PBS + Ca$^{2+}$/Mg$^{2+}$ and fixed in 70% EtOH for 10 min. Subsequently, cells were treated with MeOH for 5 min and permeabilized with 1.5 M HCl for 20 min. Cells were blocked using PBS + Ca$^{2+}$/Mg$^{2+}$, 0.5% Tween, 0.25% BSA and 5% FCS for 30 min. The cells were incubated with the primary antibodies rat-anti BrdU (Clone BU1/75, 1:20 dilution; Abcam) and mouse-monoclonal phosphorylated H2AX (Upstate, 1:100 dilutions) for 2 h and washed with PBS + Ca$^{2+}$/Mg$^{2+}$ and 0.5% Tween. Next, the cells were incubated with the secondary antibodies goat–anti Rat Alexa 568 (1:100 dilution; Invitrogen) and Goat-antimouse Alexa 488 (1:100 dilution; Invitrogen). DNA was stained using Topro3. Images were made on a confocal Leica SP5 system using a ×63 oil objective with LAS-AF software and analyzed using a customized Macro on ImageJ software.

# Supplementary Information

# Acknowledgements

We thank all members of the Te Riele lab for helpful support and discussions.

## Author Contributions

B Benedict: conceptualization, formal analysis, validation, investigation, visualization, methodology, and writing—original draft, review, and editing.

MAE van Bueren: investigation and writing—review and editing.

FPA van Gemert: investigation and writing—review and editing.

C Lieftink: formal analysis, methodology, and writing—review and editing.

S Guerrero Llobet: investigation, methodology, and writing—review and editing.

MATM van Vugt: supervision, methodology, and writing—review and editing.

RL Beijersbergen: formal analysis, supervision, methodology, and writing—review and editing.

H te Riele: conceptualization, supervision, funding acquisition, visualization, project administration, and writing—review and editing.

## Conflict of Interest Statement

The authors declare that they have no conflict of interest.

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
