## [Reviewer comments · Life Science Alliance]

Life Science Alliance

The RECQL helicase prevents replication fork collapse during replication stress

Bente Benedict, Marit van Bueren, Frank van Gemert, Cor Liefink, Sergi Guerrero Llobet, Marcel van Vugt, Roderick Beijersbergen, and Hein te Riele

DOI: <https://doi.org/10.26508/lsa.202000668>

Corresponding author(s): *Hein te Riele, Netherlands Cancer Institute*

Review Timeline:

Submission Date:	2020-02-03
Editorial Decision:	2020-02-03
Revision Received:	2020-07-03
Editorial Decision:	2020-08-03
Revision Received:	2020-08-11
Accepted:	2020-08-12

Scientific Editor: *Shachi Bhatt*

Transaction Report:

Please note that the manuscript was previously reviewed at another journal and the reports were taken into account in the decision-making process at Life Science Alliance. Since the original reviews are not subject to Life Science Alliance's transparent review process policy, the reports and author response cannot be published.

February 3, 2020

Re: Life Science Alliance manuscript #LSA-2020-00668-T

Prof. Hein te Riele
Netherlands Cancer Institute
Division of Molecular Biology
The Netherlands Cancer Institute
Plesmanlaan 121
Amsterdam NL-1066 CX
Netherlands

Dear Dr. te Riele,

Thank you for submitting your manuscript entitled "The RECQL helicase prevents replication fork collapse during replication stress" to Life Science Alliance. The manuscript was assessed by expert reviewers at another journal before, and the editors transferred those reports to us with your permission.

The reviewers who evaluated your manuscript elsewhere appreciated your work, but thought that some of your conclusions were not sufficiently supported by the data provided. We would like to invite you to submit a revised version of your manuscript based on the reviewer reports already at hand. Importantly,

- a full point-by-point response should get provided
- the reviewers point out that a direct role of RECQL in replication fork restart is not supported, and this concern should get addressed by re-wording
- the comet assay results need consolidation via a different method
- more support for DSBs being associated with nascent DNA synthesis should get provided
- issues regarding statistical tests should get addressed
- the data based on shRNA against RECQL should get confirmed with an additional shRNA to exclude off-target effects
- the concern regarding focussing on Mre11 should get addressed

The typical timeframe for revisions is three months. Please note that papers are generally considered through only one revision cycle, so strong support from the referees on the revised

version is needed for acceptance.

Thank you for this interesting contribution to Life Science Alliance. We are looking forward to receiving your revised manuscript.

Sincerely,

B. MANUSCRIPT ORGANIZATION AND FORMATTING:

August 3, 2020

RE: Life Science Alliance Manuscript #LSA-2020-00668-TRR

Prof. Hein te Riele
Netherlands Cancer Institute
Division of Molecular Biology
Plesmanlaan 121
Plesmanlaan 121
Amsterdam 1066 CX
Netherlands

Dear Dr. te Riele,

Thank you for submitting your revised manuscript entitled "The RECQL helicase prevents replication fork collapse during replication stress". We would be happy to publish your paper in Life Science Alliance pending final revisions necessary to meet our formatting guidelines.

- please add the citations pointed out by ref 1 and 3 and discussed them (the Lopes group and others)
- please add ORCID ID for corresponding author-you should have received instructions on how to do so
- please add a callout for Figure S4E to your main manuscript text
- please add scale bars to Figure 2G

A. FINAL FILES:

- An editable version of the final text (.DOC or .DOCX) is needed for copyediting (no PDFs).
- High-resolution figure, supplementary figure and video files uploaded as individual files: See our detailed guidelines for preparing your production-ready images, <http://www.life-science-alliance.org/authors>
- Summary blurb (enter in submission system): A short text summarizing in a single sentence the

study (max. 200 characters including spaces). This text is used in conjunction with the titles of papers, hence should be informative and complementary to the title. It should describe the context and significance of the findings for a general readership; it should be written in the present tense and refer to the work in the third person. Author names should not be mentioned.

B. MANUSCRIPT ORGANIZATION AND FORMATTING:

Sincerely,

Reilly Lorenz
Editorial Office Life Science Alliance
Meyerhofstr. 1
69117 Heidelberg, Germany
t +49 6221 8891 414
e contact@life-science-alliance.org
www.life-science-alliance.org

August 12, 2020

RE: Life Science Alliance Manuscript #LSA-2020-00668-TRRR

Prof. Hein te Riele
Netherlands Cancer Institute
Division of Tumor Biology and Immunology
Plesmanlaan 121
Plesmanlaan 121
Amsterdam 1066 CX
Netherlands

Dear Dr. te Riele,

Thank you for submitting your Research Article entitled "The RECQL helicase prevents replication fork collapse during replication stress". It is a pleasure to let you know that your manuscript is now accepted for publication in Life Science Alliance. Congratulations on this interesting work.

DISTRIBUTION OF MATERIALS:

Again, congratulations on a very nice paper. I hope you found the review process to be constructive and are pleased with how the manuscript was handled editorially. We look forward to future exciting submissions from your lab.

Sincerely,

Reilly Lorenz
Editorial Office Life Science Alliance
Meyerhofstr. 1
69117 Heidelberg, Germany
t +49 6221 8891 414
e contact@life-science-alliance.org
www.life-science-alliance.org